# Post-Traumatic Intraparenchymal Renal Hemorrhages: Correlation between CT and DSA Vascular Findings for Superselective Embolization Procedures

**DOI:** 10.3390/diagnostics11071256

**Published:** 2021-07-14

**Authors:** Francesco Giurazza, Andrea Contegiacomo, Fabio Corvino, Alberto Rebonato, Davide Castiglione, Stefano Palmucci, Raffaella Niola, Antonio Basile

**Affiliations:** 1Vascular and Interventional Radiology Department, Cardarelli Hospital, Via Antonio Cardarelli 9, 80131 Naples, Italy; effecorvino@gmail.com (F.C.); raffaellaniola2@gmail.com (R.N.); 2Radiology Department, Fondazione Policlinico Universitario Agostino Gemelli IRCCS, Largo Agostino Gemelli 8, 00136 Rome, Italy; andrea.contegiacomo@gmail.com; 3Radiology Department, Marche Nord Hospital, Piazzale Carlo Cinelli 1, 61121 Pesaro, Italy; alberto.rebonato@ospedalimarchenord.it; 4Radiology 1 Unit, Dipartimento di Scienze Mediche, Chirurgiche e Tecnologie Avanzate, Università degli Studi di Catania, Via S. Sofia 98, 95100 Catania, Italy; dcastiglione@unict.it (D.C.); spalmucci@unict.it (S.P.); basile.antonello73@gmail.com (A.B.)

**Keywords:** renal hemorrhage, intraparenchymal, trauma, computed tomography, superselective embolization

## Abstract

Background: This study aims to investigate the correlation between computed tomography (CT) and digital subtraction angiography (DSA) findings in patients affected by acute post-traumatic intraparenchymal renal hemorrhages and evaluate their conservative management with superselective embolization. Methods: This retrospective multicenter analysis focuses on patients affected by renal bleedings detected by contrast-enhanced CT and treated with superselective endovascular embolization. CT findings were compared to DSA. Embolization procedural data were analyzed and renal function was evaluated before and after the intervention. Results: Twenty-seven patients were retrospectively evaluated in one year. Compared to DSA, CT showed 96.3% diagnostic accuracy in terms of hemorrhage recognition; concerning the type of vascular lesion, there was discrepancy between CT and DSA in five cases. The technical success rate of embolization was 100%, while primary clinical success was 88.9%. The inferior parenchymal third was the most frequent site of renal injury. Microcoils were the most adopted embolics. Renal function did not change significantly before and after embolization. Conclusions: CT has elevated diagnostic accuracy in detecting post-traumatic intraparenchymal renal hemorrhages; in a small percentage, the type of vascular lesion may differ from the findings observed at DSA. In this scenario, superselective embolization presents high clinical success with a low complication rate.

## 1. Introduction

Kidneys are positioned in the retroperitoneal space and are surrounded by adipose tissue; this anatomical “protected” area makes kidney the third solid organ involved in abdominal blunt or penetrating trauma in terms of frequency, after liver and spleen [1]. Injury of the genitourinary tract occurs in 10% of all abdominal traumas [2], and the kidney is involved 65–90% of the time [3].

According to the World Society of Emergency Surgery (WSES) guidelines, the first evaluation must assess the hemodynamic status: unstable patients require emergent surgery while stable patients can be managed with non-operative management, including embolization [3].

Together with clinical evaluation, imaging has an essential role in patient management. With the technological progress in the last two decades, CT with intravenous contrast has evolved into a central role in the evaluation of patients with renal trauma; with a single bolus of intravenous contrast, thin-section CT images can be rapidly acquired that are highly sensitive and specific for renal injuries [4]. At the same time, CT is an essential tool for interventional radiologists to plan the embolization procedure: multiplanar reconstructions (MPR) and maximum intensity projections (MIP) can be applied to create vascular maps and visualize the type of vascular lesion and the vessels involved [5].

In almost 90% of all renal traumas, patients present with hemodynamic stability; therefore, non-operative management is largely applied in this scenario: if computed tomography (CT) identifies an active arterial bleeding, embolization is indicated. Therefore, embolization presents a relevant role in renal trauma thanks to its efficacy and the possibility of sparing healthy renal parenchyma [6].

Renal artery embolization was first described by Almgard in 1973 [7]; it was defined as the voluntary occlusion of the renal artery and/or one or more of its branches by means of definitive or temporary embolic agents injected through an endovascular catheter [8]. The advent of new materials and embolic agents, as well as the development of the technique itself, have extended the indications of renal artery embolization over time, making this procedure an accepted and minimally invasive alternative to surgery in the management of clinically stable patients with renal vascular injuries [6].

This study aims to investigate the correlation between CT and digital subtraction angiography (DSA) findings in patients affected by acute post-traumatic intraparenchymal renal hemorrhages and evaluate their conservative management with superselective embolization in terms of technical and clinical successes.

## 2. Materials and Methods

This is a retrospective multicenter analysis; local electronical databases and picture archiving systems were examined, considering the year 2020.

All patients signed an informed consent before the procedure, when the clinical conditions allowed.

### 2.1. Sample

Patients affected by renal bleeding and treated with endovascular embolization were considered.

Inclusion criteria were: traumatic etiology, intraparenchymal bleeding lesions, availability of pre-procedural contrast-enhanced CT scan, superselective embolization, pre- and postprocedural Creatinine values (almost two weeks after embolization).

Exclusion criteria were: extraparenchymal vascular lesions, surgery performed before embolization, non-selective embolization, COVID-19 infection and age <18 years.

All patients presented with the three classic symptoms of renal trauma: flank pain in correspondence of the affected kidney, gross hematuria and anemia; no patients had signs of hemodynamic shock (systolic pressure <90 mmHg).

Renal function was evaluated before and after embolization, with a follow-up at almost two weeks, considering the glomerular filtration rate estimated using the Modification of Diet in Renal Diseases (MDRD) formula [9].

### 2.2. CT Scan Protocol

CT scans of the whole abdomen were acquired according to a standardized protocol; 128-slices multidetector CT (MDCT) scanners were adopted (GE Revolution GSI, General Electric Company^®^, Little Chalfont, UK; SOMATOM Definition Flash, Siemens Healthcare^®^, Erlangen, Germany), with a slice thickness of 0.6 mm. Patients were positioned supine with the arms abducted when feasible.

First, a non-enhanced CT scan was acquired. An automated bolus tracking, with the region of interest placed into the aortic arch at an attenuation threshold of 100 HU, was used to time the beginning of the arterial phase. Venous, delayed and excretory phases were performed at a 60–70, 180 and 420 s delay from the end of the injection, respectively.

The intravenous iodinated contrast agent consisted of 100–120 mL (Iomeron 400; Bracco Diagnostics^®^, Milan, Italy; Ultravist 370; Bayer^®^, Leverkusen, Germany), followed by 40 mL of saline chaser; the flow rate was 4 mL/s.

All images have been evaluated using post-processing protocols, MPR and MIP.

Site of bleeding was evaluated, considering the renal parenchyma in three portions: inferior, middle, superior third.

The type of vascular lesion was recorded. Pseudoaneurysms (Figure 1) were defined as focal round or oval irregularity of the outer contour of the arterial vessel wall of equal density to the intra-arterial contrast material, with a rim of separation from the vessel lumen.

Arteriovenous fistula (Figure 2) consisted of a traumatic communication between the arterial and venous systems, seen as an asymmetrical, early contrast opacification of a vein, during the arterial phase.

Active bleeding (Figure 3) was recognized as a region of extraluminal contrast agent extravasation, which increases and changes its morphology during multiphasic acquisitions [6,10].

Concomitant perirenal hematoma was investigated if associated with a specific type of vascular injury.

CT diagnostic accuracy was analyzed with concordance between CT and DSA findings, considering the latter as the gold standard of reference.

### 2.3. Embolization Procedure

Procedures were conducted by interventional radiologists with almost 5 years of experience in the embolization field.

The angiosuites were equipped with Infinix I 8000 C Toshiba^®^ Tokyo, Japan; Azurion Philips^®^, Amsterdam, The Netherlands; Artis Zee C-arms Siemens^®^, Erlangen, Germany

Interventions were performed in emergent settings after traumatic lesions involving intraparenchymal branches of the renal artery. The coronal MIP CT scan reconstructions were carefully analyzed in order to plan the procedure. The length of time between CT scan acquisition and embolization was recorded.

A standard technique was applied: under ultrasound guidance, the right common femoral artery was punctured, and a 11 cm 5 French (Fr) introducer was positioned. Renal DSA was acquired using a 5 Fr diagnostic catheter (Cobra 1, Simmons 1 or Shepherd’s tips according to the vascular anatomy) to confirm CT findings. After selection of the proper working projection, a 2.7 or 2.8 Fr microcatheter (Progreat, Terumo^®^, Shibuya, Japan) was advanced distally in correspondence of the intraparenchymal bleeding site; embolic agents were superselectively released to spare the largest portion of healthy renal parenchyma. A final DSA was acquired from the 5 Fr diagnostic catheter to record the technical success or detect remnant site of bleedings requiring further embolization.

The puncture access was sealed using a mechanical closure device (Angioseal 6VIP or Femoseal, Terumo^®^, Shibuya, Japan) and a compressive bandage. Patients remained supine 24 h, when femoral bandage was removed.

All interventions were performed under local anesthesia and sedo-analgesia.

Wide spectrum antibiotics were administered.

Interval time between CT and interventional procedure was recorded.

Technical success was considered as complete disappearance of the bleeding foci at the last DSA run with preserved flow in the adjacent renal branches.

Clinical success was considered as improvement in hemoglobin values, resolution/relief in flank pain and disappearance of gross hematuria.

### 2.4. Statistical Analysis

Statistical analysis was performed with a dedicated software (SPSS 22.0, IBM ^®^, Armonk, NY, USA).

The Kolmogorov–Smirnov test of normality was employed to assess normal distribution of continuous variables. Student *t*-test was used for continuous variables analysis.

Ordinal and categorical variables were expressed as frequencies and percentages, and Chi-square test was adopted for their analysis.

## 3. Results

### 3.1. Population Data

Twenty-seven patients were included in this study (Table 1); there were eight females and 19 males; mean age was 52.74 years (range: 19–85).

In 24 patients, bleeding had iatrogenic etiology, while in three subjects it occurred after driving accident. Among the iatrogenic lesions, 21 were secondary to percutaneous urological procedures, such as percutaneous nephrostomy or percutaneous nephrolithotomy; two occurred after percutaneous biopsy and one after renal transplantation.

No patient was under antiplatelet or anticoagulation therapy.

### 3.2. CT-DSA Correlation

Time between CT acquisition and DSA had a mean of 0.52 days (range: 0–4) (Table 2).

At CT, 13 patients presented with a pseudoaneurysm, 12 with contrast agent extravasation and one with an arteriovenous fistula; one patient had no sign of active hemorrhage and so CT was negative.

At DSA, nine patients presented with active bleeding, 15 had a pseudoaneurysm, one had an arteriovenous fistula, one had both active bleeding and pseudoaneurysm while one had both active bleeding and arterio-venous fistula.

Compared to DSA, CT showed 96.3% diagnostic accuracy in terms of hemorrhage recognition: 26 out of 27 patients showed signs of active bleeding. Concerning the type of vascular lesion, in 5 (19.2%) of the 26 positive subjects at CT, there was a discrepancy with DSA: in two patients, CT detected an active bleeding, while at DSA there was a pseudoaneurysm; in one case CT detected a pseudoaneurysm, while at DSA there was an active bleeding; in two subjects, CT detected an active bleeding while at DSA there were also a pseudoaneurysm and a fistula. In all five cases, embolization was performed >2 days after CT acquisition.

### 3.3. Embolization Data

Technical success was obtained in all cases (100%) (Table 3).

Primary clinical success was reached in 88.9% of cases: in three patients (11.1%), the embolization was repeated because of coils’ recanalization, with secondary clinical success obtained in all subjects (100%).

The intraparenchymal bleeding was located in the inferior third in 13 patients, in the middle portion in 8 and in the superior third in 6.

Microcoils, both pushable (Tornado or MicroNester, Cook Medical^®^, Bloomington, IN, USA) and detachable (Concerto Helix, ev3-Medtronic^®^, Minneapolis, MN, USA), were the embolic agent adopted most often; they were used as the sole embolic agent in 17 patients. In three cases, they were associated with a MicroVascular Plug (MVP-5Q, Reverse Medical-Medtronic^®^, Minneapolis, MN, USA), and in one case they were associated with microparticles (Embosphere 300–500 micron, Merit Medical^®^, South Jordan, UT, USA). The MicroVascular Plug (Reverse Medical-Medtronic^®^, Minneapolis, MN, USA) was adopted as the sole embolic agent in four patients: MVP-3Q and MVP-5Q were chosen in two cases each. Microparticles (Embosphere 300–500 micron, Merit Medical^®^, South Jordan, UT, USA) and cohesive liquid embolic (Phil 30%, Microvention^®^, Aliso Viejo, CA, USA) were applied once each.

Perirenal hematoma was observed in 17 patients (63%): no differences were observed in terms of lesion type and presence/absence of perirenal hematoma (*p* = 0.27).

The renal function did not worsen after the embolization: pre-procedural and post-procedural GFR values were normally distributed and did not differ significantly (*p*-value = 0.37); pre- and post-procedural mean GFR values were 67.1 mL/min (range: 7.9–146.1) and 80.3 mL/min (range: 10.4–171.7), respectively.

According to the CIRSE classification system [11], only two grade I complications occurred (7.4%); both were hematoma at the site of the femoral vascular access, not requiring further intervention.

## 4. Discussion

Contrast-enhanced CT presents high diagnostic accuracy in the detection of renal hemorrhage [12]. In this series, pseudoaneurysms and active bleeding were the most frequent type of vascular lesion; CT showed 96.3% diagnostic accuracy in terms of hemorrhage recognition, similar to previous studies [13]. One patient was embolized even if CT was negative, because they presented with massive perirenal hematoma and contrast agent extravasation only in the delayed acquisition phase, despite a sudden fall in hemoglobin value. At DSA, this patient showed an active extraluminal bleeding, which was probably not detected by CT because of hematoma compression.

Perirenal hematoma was not related to a specific vascular injury.

In five patients (19.2%), differences were noted between the type of vascular lesion recognized at CT and DSA. However, from a management perspective, this represents a minor issue because, in the patient selection, the indication for intraparenchymal renal embolization is based on the presence of the hemorrhagic lesion and not on the type of lesion. Furthermore, in all five patients, DSA was not conducted on the same day as CT acquisition, and this might influence this value in terms of possible evolution of the parenchymal damage. Ideally, CT and embolization should be performed within the shortest possible timeframe.

Concerning the procedural planning, CT is mandatory in the arterial phase: thanks to MPR and MIP algorithms, this allows the vascular map that interventional radiologists will face during the procedure to reach the target area to be forecast. In addition, the endovascular equipment (guidewire, catheters, microcatheters etc.) required to accomplish the intervention can be predicted. As well as reducing technical inconveniences, this approach can decrease the duration of the intervention, with benefits in terms of blood loss and X-ray dose.

In this series, the main cause of bleeding was iatrogenic; this explains why lesions mainly occurred in the inferior third of the renal parenchyma, which is the target site for percutaneous nephrostomic access [14]. Furthermore, laparoscopic and percutaneous urological interventions (as laparoscopic nephrectomy, percutaneous nephrolithotomy, nephrostomy and biopsy) are increasingly performed by urologists to reduce parenchymal loss compared to open surgery [15]; however, this has increased the rate of life-threatening arterial injuries [16,17].

In this regard, the primary clinical success rate was high (88.9%), confirming the experiences of previous authors, which have already demonstrated the efficacy of superselective embolization. Pozzi Mucelli et al. [2] reported on the effectiveness of embolization above 85% in renal trauma; Contegiacomo et al. [15] reviewed 5 years of renal artery embolization for iatrogenic renal vascular injuries, describing clinical success after a single embolization of 89.3%; Giurazza et al. [18,19] focused their analysis on the use of the MVP in renal hemorrhages and reported 100% clinical successes, with the preservation of healthy renal parenchima; Mahdjoub et al. [20] analyzed the effectiveness of liquid embolization with ethylene vinyl alcohol copolymer, reporting 100% technical and clinical success.

The value of embolization derives from the possibility of simultaneous hemorrhage interruption and healthy parenchyma preservation; from the clinical perspective, this aspect is of paramount importance because the renal function is maintained.

No statistically significant differences were noted in this sample between GFR before and after embolization. This is in accordance with the literature: Haochen et al. reported a mean creatinine value that was not different at 1 day and 1 week after embolization compared with before embolization [21]; similarly, Dong et al. did not find significant differences between serum creatinine before and 7 days after superselective renal arterial embolization) [22]. On the other hand, Lee et al. found chronic kidney disease and a high embolization grade to be predictive factors for postprocedural acute kidney injuries [23].

Regarding procedural complications, the rate reported here is low and confirms the intervention’s safety, as described in previous study: Giurazza et al. found no complications according to CIRSE classification standards [18]. Neither did Dong et al., who saw no major complications in 49 patients [22]; Mahdjoub et al. also reported no serious complications attributable to liquid embolization, but found a mean infarcted renal area of 10% [20].

In the last decade, technological progress has produced embolic devices that allow interventional radiologists to be highly precise during embolization procedures, reducing the risk of non-target embolizations. In this sample, the most commonly adopted devices were mechanical agents (microcoils and microvascular plugs) that can be released distally into the renal parenchyma through a microcatheter; this allows for the avoidance of proximal embolization, sparing healthy parenchyma. Compared to particulated and liquid agents, mechanical embolics, especially if detachable, provide better control. In the literature, most of the series report coils as the embolic agent of choice for renal hemorrhage in various clinical settings. Guo et al. obtained 81.5% hemostasis with coils alone in 27 patients [24]; Sam et al. adopted microcoils alone or in combination with particles, gelfoam and glue in 44 of 50 patients, obtaining 96% clinical success [25]; Wang et al. used coils alone in 40 of 46 patients with 80.8% success [26]; Ierardi et al. chose coils in 19 of 21 patients [27], reporting 100% technical success and 95% clinical success. However, to the best of our knowledge, no studies have specifically compared the different embolics, and there are few findings concerning the use of glue [15,28]. In this sample, one patient was treated with Phil^®^, which is part of the relatively novel cohesive liquid embolics, which are mainly adopted for the treatment of vascular malformations [29,30] but seem to also be effective for renal hemorrhages [20].

This study presents some limitations, such as its retrospective nature and the small number of patients; however, this is considered a urgent scenario and a prospective design is difficult to obtain. Furthermore, the small number of patients analyzed is related to the limited lapse of time considered, of only patients treated during the year 2020: this was chosen in order to avoid possible bias related to technological progress and improvements in the operators’ learning curve.

## 5. Conclusions

This experience confirms that contrast-enhanced CT presents an elevated accuracy in detecting intraparenchymal renal hemorrhages; in a minor number of cases, the type of vascular lesion at CT is different from the DSA findings. Superselective embolization is able to control renal bleedings with high clinical success, preserving the renal function, and has a low complications rate. In this series, microcoils were the most frequently used embolic agents. Future studies are needed to confirm the findings of this paper, with a larger population, comparing the different embolic agents.

## Figures and Tables

**Figure 1 diagnostics-11-01256-f001:**
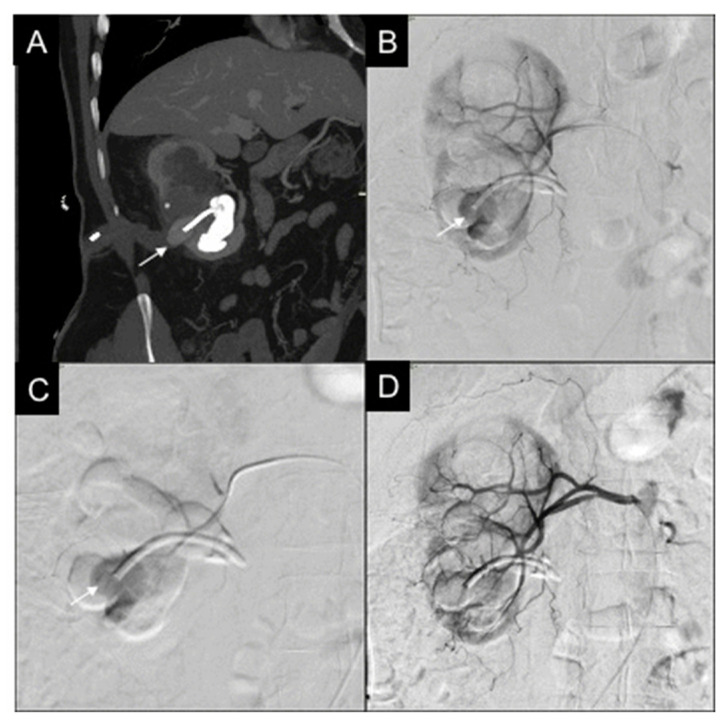
46-year-old male, affected by iatrogenic PA. (**A**) CT reconstruction with MIP and coronal oblique MPR, in venous phase, shows the PA (white arrow) in correspondence with the point of entrance of the nephrostomy into the renal cortex; amount of intracaliceal contrast agent residual from the urological procedure. (**B**) Selective right renal DSA confirms PA (white arrow). (**C**) Microcatheter positioning into the feeder of the PA (white arrow). (**D**) Final DSA showing PA resolution after embolization with a microvascular plug (MVP-5Q), renal parenchyma normally vascularized.

**Figure 2 diagnostics-11-01256-f002:**
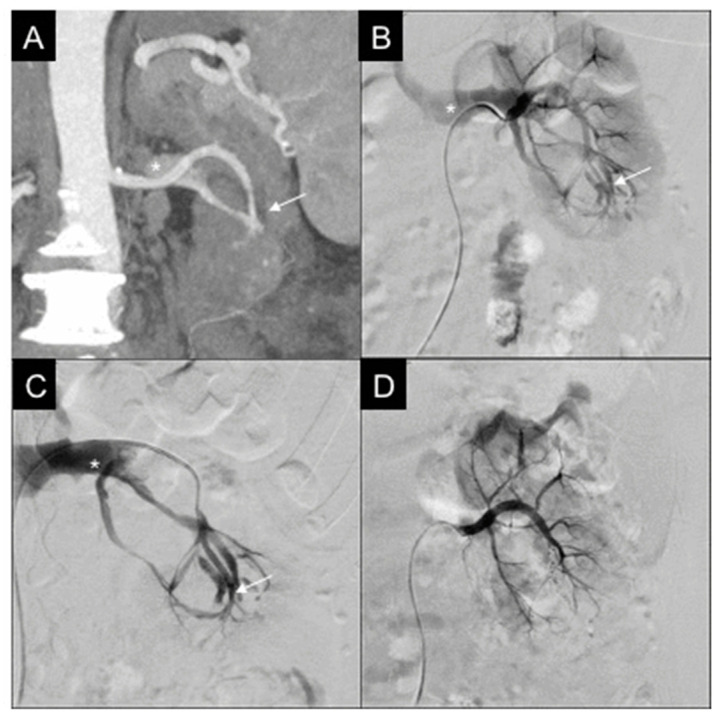
57-year-old female, affected by iatrogenic arteriovenous fistula (AVF). (**A**) CT reconstruction with MIP and coronal MPR, in arterial phase: intraparenchymal AVF (white arrow) at the inferior third of the left kidney, with early opacification of the left renal vein (white asterisk). (**B**) Selective right renal DSA confirms AVF (white arrow), left renal vein opacified in early phase of injection (white asterisk). (**C**) Selective catheterization of the inferior third, up to the AVF (white arrow) with evidence of the renal vein (white asterisk). (**D**) Final DSA showing AVF resolution after embolization with two pushable microcoils (MicroNester 4 mm), renal parenchyma being vascularized, renal vein no longer opacified.

**Figure 3 diagnostics-11-01256-f003:**
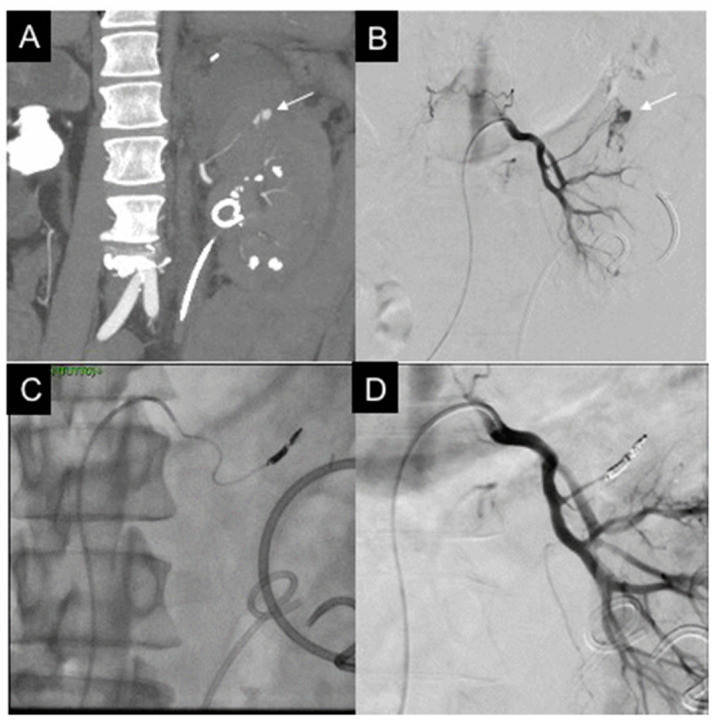
50-year-old male, affected by iatrogenic active bleeding. (**A**) CT reconstruction with MIP and coronal MPR, in arterial phase: intraparenchymal active bleeding (white arrow) at the superior third of the left kidney; homolateral urological stent is evident. (**B**) Selective left renal DSA confirms active bleeding (white arrow). (**C**) Superselective microcatheterization and embolization with two detachable coils (Concerto 4 mm × 8 cm). (**D**) Final DSA showing active bleeding resolution, renal parenchyma vascularized.

**Table 1 diagnostics-11-01256-t001:** Sample data.

Age	52.74 (Range: 19–85)
Sex	8 females
	19 males
Trauma etiology	24 iatrogenic
	3 driving
CT to DSA interval	0.52 days (range: 0–4)
Target area	13 inferior third
	8 middle third
	6 superior third

CT: computed tomography; DSA: digital subtraction angiography.

**Table 2 diagnostics-11-01256-t002:** CT and DSA data.

CT to DSA Interval	0.52 Days (Range: 0–4)
Target area	13 inferior third
	8 middle third
	6 superior third
Type of lesion at CT	13 PA
	12 active blushes
	1 AVF
	1 negative
Type of lesion at DSA	15 PA
	9 active blushes
	1 AVF
	1 active blush + PA
	1 active blush + AVF
Perirenal hematoma	17 yes
	10 no

CT: computed tomography; DSA: digital subtraction angiography; PA: pseudoaneurysm; AVF: arteriovenous fistula.

**Table 3 diagnostics-11-01256-t003:** Embolization data.

Embolic agent	16 microcoils
	3 microcoils + MVP
	1 microcoils + particles
	4 MVP
	1 particles
	1 liquid agent (Phil)
Preprocedural GFR (mL/min)	67.1 (7.9–146.1)
Postprocedural GFR (mL/min)	80.3 (range: 10.4–171.7)
Technical success	100%
Clinical success	88.9% primary
	100% secondary

GFR: glomerular filtration rate; ml: milliltier; min: minute; MVP: Microvascular plug.

## Data Availability

The data presented in this study are available on request from the corresponding author (francescogiurazza@hotmail.it). The data are not publicly available for the privacy protection of the patients involved, in accordance with the current law of the country in which the study was conducted.

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
