# Peer review of "Post-Traumatic Intraparenchymal Renal Hemorrhages: Correlation between CT and DSA Vascular Findings for Superselective Embolization Procedures"

_diagnostics, 2021, doi:10.3390/diagnostics11071256_

Round 1

Reviewer 1 Report

Dear All,

The manuscript: “Post-traumatic intraparenchymal renal hemorrhages: correlation between CT and DSA vascular findings for superselective embolization procedures” by Giurazza et al. investigated the correlation between computed tomography and digital subtraction angiography in 27 patients affected by acute post-traumatic intraparenchymal renal hemorrhages and evaluate their conservative management with superselective embolization.

The article is very interesting, showing promising results for the field.

Please, take into consideration the following major remarks:

  1. The introduction is too short. You should add more paragraphs with more data about: post-traumatic intraparenchymal renal hemorrhages, CT investigation in renal hemorrhages, DSA technique, DSA in renal hemorrhages.

Also, there are some missing references in the introduction (after lines 52-55 and after lines 56-59).

  1. The titles of the figures must be placed below them, not above them. Moreover, try to simplify as much as you can their description from the title.
  2. The quality of tables is poor. Please try to improve them.
  3. The manuscript needs a proofreading to correct the typographical and grammatical errors.

Reviewer 2 Report

The introductory section should be developed and more recent references should be added in this section. 

Results section, raw 215: "Compared to DSA, CT showed...". What was the diagnostic accuracy of DSA in your study? 

Discussions section: there should not be used so many refereces at the end of a single phrase, like raw 287 [2,12,13,15-20]; raw 290 [12, 14-20]; raw 292 [2, 12, 14-20] and raw 302 [21-25]. Rephrase and develop the text so that references would be indexed to specific fragments of text. 

English language editing is needed. 

Round 2

Reviewer 1 Report

The manuscript: “Post-traumatic intraparenchymal renal hemorrhages: correlation between CT and DSA vascular findings for superselective embolization procedures” by Giurazza et al. investigated the correlation between computed tomography and digital subtraction angiography in 27 patients affected by acute post-traumatic intraparenchymal renal hemorrhages and evaluate their conservative management with superselective embolization.

The article is very interesting, showing promising results for the field.

Please, take into consideration the following major remarks:

  1. The introduction is too short. You should add more paragraphs with more data about: post-traumatic intraparenchymal renal hemorrhages, CT investigation in renal hemorrhages, DSA technique, DSA in renal hemorrhages.

Also, there are some missing references in the introduction (after lines 52-55 and after lines 56-59).

  1. The titles of the figures must be placed below them, not above them. Moreover, try to simplify as much as you can their description from the title.
  2. The quality of tables is poor. Please try to improve them.
  3. The manuscript needs a proofreading to correct the typographical and grammatical errors.

Author Response

Dear Reviewer 1,

first of all thank you for your precious time spended in reviewing our paper, especially in this COVID era.

We have carefully read your kind observations and we have answered point by point, as you can see below.

We hope to have ameliorated the manuscript as suggested.

We really care about this manuscript to be published because we have made great efforts to provide this multicenter experience.

Our bests

Francesco Giurazza, MD-PhD-EBIR and co-authors

Observation 1: Moderate English changes required

Answer: We have improved English language with the help of a native English mother-tongue speaker.

Observation 2: Does the introduction provide sufficient background and include all relevant references? Must be improved

Answer: We agree with the observation; please see also with answer to observation 6.

Observation 3: The introduction is too short. You should add more paragraphs with more data about: post-traumatic intraparenchymal renal hemorrhages, CT investigation in renal hemorrhages, DSA technique, DSA in renal hemorrhages.

Answer: We agree with this observation, therefore we have added the following paragraphs to the introduction section, inserting also new references:

“With advances in CT technology in the past two decades, CT with intravenous contrast has evolved into a central role in the evaluation of patients with renal trauma; with a single bolus of intravenous contrast, thin-section CT images can be rapidly acquired that are highly sensitive and specific for renal injuries [4].”

and

“Renal artery embolization was first described by Almgard in 1973 [7]; it was defined as the voluntary occlusion of the renal artery and/or one or more of its branches, by means of definitive or temporary embolic agents injected through an endovascular catheter [8] The advent of new materials and embolic agents, as well as the development of the technique itself, have extended the indications of renal artery embolization over time, making this procedure an accepted and minimally invasive alternative to surgery in the management of clinically stable patients with renal vascular injuries.”

Observation 4: Also, there are some missing references in the introduction (after lines 52-55 and after lines 56-59).

Answer: We agree with this observation; therefore we have included the references where suggested. Please see the references highlighted in blue; we have modified the references list as suggested.

Observation 5: The titles of the figures must be placed below them, not above them. Moreover, try to simplify as much as you can their description from the title.

Answer: We agree with this observation, therefore we have positioned the titles of the figures below the figures and we have simplified their description.

Observation 6: The quality of tables is poor. Please try to improve them.

Answer: We agree with this observation, therefore we have modified the tables improving the schematic quality; please Tables file.

Observation 7: The manuscript needs a proofreading to correct the typographical and grammatical errors.

Answer: See answer to observation 1.

Reviewer 2 Report

The authors addressed all the issues from the first review, thus the quality of the manuscript significantly improved. I recommend publication. 

Author Response

Dear reviewer 2,

Many thanks for your comments

Round 3

Reviewer 1 Report

The authors have taken into account all my suggestions. I consider that the changes have improved the quality of the article and can be accepted for publication in  present form.